# Study on Shear Strength of Soil–Root Systems of Different Vegetation Types

**DOI:** 10.3390/plants13212963

**Published:** 2024-10-23

**Authors:** Shengqi Jian, Yixue Niu, Xueli Zhang, Yi Wei, Jun Gao

**Affiliations:** College of Water Conservancy and Transportation, Zhengzhou University, Zhengzhou 450001, China; jiansq@zzu.edu.cn (S.J.); nyxue1208@163.com (Y.N.); weiyi155@163.com (Y.W.); gj2suanru@163.com (J.G.)

**Keywords:** slope stability, root morphological, root tensile strength, in situ shear test, soil reinforcement

## Abstract

The root systems of vegetation significantly contribute to enhancing slope stability. The shear strength of soil–root systems is a crucial parameter for assessing slope stability. This study focuses on six types of vegetation in the Yellow River Basin of China (woodland: *Populus przewalskii* and *Broussonetia papyrifera*; shrubland: *Periploca sepium* and *Ziziphus jujuba*; grassland: *Artemisia hedinii* and *Setaria viridis*), employing in situ shear tests and the Wu–Waldron model (Wu model) to investigate the shear strength of soil–root systems. The results show that the shear stress–displacement curves for *P. przewalskii*, *B. papyrifera*, and *Z. jujuba* are higher and steeper, with clear inflection points. The tensile strength of the roots from the six vegetation types decreases as the root diameter increases. According to the Wu model, the additional root cohesion is ranked as follows: *A. hedinii* > *B. papyrifera* > *P. przewalskii* > *Z. jujuba* > *P. sepium* > *S. viridis*. Based on the in situ shear tests, the shear strength increments are ranked as follows: *Z. jujuba* > *B. papyrifera* > *P. przewalskii* > *A. hedinii* > *P. sepium* > *S. viridis*. Overall, the additional root cohesion obtained by the Wu model in each soil layer is greater than the shear strength increment measured from the in situ shear tests. In the 0–30 cm soil layers, the soil–root systems of *Z. jujuba*, *B. papyrifera*, *P. przewalskii*, and *A. hedinii* exhibit a better shear strength, whereas *P. sepium* and *S. viridis* perform poorly. A principal component analysis reveals that the shear strength of the soil–root systems of different vegetation types is primarily influenced by the soil moisture content and root mass density. *Z. jujuba*, *B. papyrifera*, *P. przewalskii*, and *A. hedinii* are recommended for ecological restoration projects in the Yellow River Basin of China.

## 1. Introduction

To control soil erosion and slope instability, the Chinese government has implemented a series of ecological protection and restoration measures over the past few decades, including sprayed concrete layers, anti-slide retaining walls, anti-slide piles, and anchors. Although these traditional engineering protections can reinforce slopes to some extent, they cause irreversible damage to the ecological environment, fail to balance slope protection with ecological preservation, and are difficult to maintain in the long term. In recent years, the contribution of vegetation roots to slope reinforcement, due to the friction and interlocking effects between roots and soil [1], has received increasing attention [2,3,4]. Vegetation roots enhance the slope stability by improving the soil permeability, aggregate stability, and shear strength. Plant roots have strong tensile strength but weak compressive strength, whereas soil exhibits the opposite characteristics [5]. The soil–root system combines the advantages of both soil and roots, converting the shear force on the soil into tensile force in the roots through the friction between soil particles and root surfaces [6]. This mechanical reinforcement increases the soil’s shear strength, thereby mitigating soil erosion and water loss [7,8]. The role of roots in improving slope stability has long been recognized [9]. The technique of using vegetation to stabilize slopes and control soil erosion has been widely accepted in many countries and regions [10].

Currently, methods for quantifying the shear strength of vegetation soil–root systems mainly include experimental methods (laboratory shear tests, in situ shear tests, root tensile tests, etc.) [11,12] and modeling methods (the Wu–Waldron Model, fiber bundle model, root bundle model, etc.) [13,14,15]. The in situ shear test is widely regarded as a reliable method because it does not disturb the soil, thereby better reflecting natural conditions [16,17]. The in situ shear test can obtain the shear stress–displacement curves of different soil–root systems. The shear strength increment of the soil–root system indicates the increase in shear strength due to the presence of roots compared to the shear strength of fallow land, reflecting the slope protection capability of different vegetation types. Xing et al. [6] conducted in situ shear tests using 25 cm × 25 cm × 10 cm shear boxes to study the effects of land use types on the strain energy, peak shear stress, and peak shear stress–displacement. Comino and Druetta [18] performed in situ shear tests in the alpine environment of Italy, showing that the presence of roots increased the shear strength of alpine soils. The Wu–Waldron model (Wu model) is easy to understand, and its parameters can be easily obtained through experiments, making it commonly used in vegetation slope protection research [19]. However, this model does not consider progressive failure due to varying displacements and individual root morphology [11], thus overestimating the effect of root reinforcement on the soil shear stress [20,21]. De Baets et al. [9] used the Wu model to quantify and rank the soil reinforcement effects of 25 typical Mediterranean plants. Su et al. [19] applied the Wu model to investigate root reinforcement in the Loess Plateau after reforestation, estimating its impact on the root mechanical properties.

Plant roots can promote the formation of soil aggregates, reduce the soil bulk density, and increase the soil cohesion and organic matter content [22,23], thereby enhancing the soil shear strength. The root length density, root surface area density, and root volume density are morphological indicators reflecting the impact of plant roots on the soil shear strength, whereas the root mass density indicates the effect of the root weight on the soil shear strength. The soil organic matter content affects the soil aggregate stability [24,25,26]. The shear strength of soil–root systems is directly related to the soil bulk density, cohesion, impermeability, soil moisture content, aggregate stability, organic matter content, and other properties [27,28,29]. Generally, the shear strength increases with soil compaction [30]. Reubens et al. [31] summarized the root characteristics affecting soil erosion and mass movement, including the root density, root length density, root number, and root area ratio. Therefore, root characteristics and soil properties are key indicators influencing the mechanical behavior of root–soil complexes.

The Huguo watershed is located in the hilly region of the Yellow River Basin. Due to its steep terrain, uneven rainfall, and the impact of human activities, soil erosion in the loess hilly area is quite severe, posing a significant threat to the local ecosystem. Therefore, selecting this area as the research object to explore the impact of vegetation on slope protection will provide important theoretical support and practical guidance for improving the ecological environment of the Yellow River Basin. Su et al. [19] explored the root characteristics of herbaceous plants, root–soil shear stress, and their influencing factors in the loess hilly region. They found that over 65% of the roots of herbaceous plants are distributed in the 0–20 cm soil layer, and the plant roots enhance the peak shear stress of the soil. This study focuses on six typical vegetation types in the Huguo watershed of the Yiluo River Basin in the Yellow River region, including woodland, shrubland, and grassland. It investigates the shear strength of soil–root systems of different vegetation types based on in situ shear tests and root–soil mechanical analysis models. The study aims to (1) obtain the shear stress–displacement curves of soil–root systems in the presence of roots through in situ shear tests on six plant species; (2) quantify the additional root cohesion for six vegetation types using the Wu model; (3) analyze the relationship between shear strength, soil properties, and root traits in in situ shear tests. The research results can provide a scientific basis for optimizing vegetation for slope protection.

## 2. Materials and Methods

### 2.1. Study Area

The Hugou Basin (34°12′–34°16′ N, 112°03′–112°08′ E) is part of the Yi River system within the Yellow River Basin in China. It represents the soil erosion characteristics of the hilly and gully regions of western Henan Province. The basin covers an area of 13.34 km^2^, with an elevation range of 332.7 to 737.0 m. It features a warm temperate continental monsoon climate, with an average annual temperature of 14 °C and an average annual precipitation of 690 mm. The upper part of the Hugou Basin is a rocky mountainous area, while the lower part belongs to the third sub-region of the loess hilly and gully area. The rocky mountainous area contains a large amount of igneous rock and a small amount of sedimentary rock, while the loess hilly area is covered by loess-like parent material. The soil particles are primarily brown earth, with a small amount of alluvial soil. The basin has a rich diversity of plant species, with 2.15 km^2^ of woodland and shrubland coverage and 1.36 km^2^ of grassland coverage.

In May 2023, a field survey was conducted in the Hugou Basin. A southwest-facing slope with an average gradient of 10° was selected as the experimental plot (30 m × 30 m). Six dominant vegetation types within the plot were selected for in situ field tests conducted from May to October 2023 (Figure 1c). For each vegetation type, 20–30 individual plants were selected for measurements (Table 1). Based on these measurements, 12 standard plants of each vegetation type were selected for in situ shear tests and root tensile tests (6 for shear tests and 6 for tensile tests). Weather conditions were stable during the shear tests, with no rainfall occurring within 10 days prior to each test. Tests were conducted daily between 8:00 and 10:00 a.m. to minimize the influence of environmental factors on the results.

*Artemisia hedinii* (*A. hedinii*): An annual herb of the Asteraceae family, featuring a single, vertical root and strong adaptability.

*Setaria viridis* (*S. viridis*): An annual herb of the Poaceae family, predominantly featuring horizontally distributed roots with well-developed fibrous roots in the upper part.

*Periploca sepium* (*P. sepium*): A species of the Apocynaceae family, capable of forming clusters after planting. It has strong adaptability and is beneficial for sand fixation, as well as soil and water conservation.

*Ziziphus jujuba* (*Z. jujuba*): A species of the Rhamnaceae family, with deep and extensive root distribution, strong resistance to cold, drought, and poor soil conditions.

*Populus przewalskii* (*P. przewalskii*): A species of the Salicaceae family, characterized by a straight trunk, smooth or vertically cracked bark, often gray-white in color, with predominantly horizontal roots and fewer vertical roots.

*Broussonetia papyrifera* (*B. papyrifera*): A species of the Moraceae family, light-loving, with strong adaptability, drought and barren tolerance, fast-growing roots, and high medicinal and economic value.

### 2.2. Measurement of Soil Properties and Root Traits

The bulk density (*BD*, g∙cm^−3^), soil water content (*SWC*, % weight), and organic matter content (*SOM*, g∙kg^−1^) of soil layers at depths of 0–10, 10–20, 20–30, 30–40, and 40–50 cm were measured for the six vegetation types. Soil bulk density and soil water content were measured using a ring knife with an inner diameter of 5 cm, and soil organic matter content was determined using the potassium dichromate volumetric method [32].

After collecting the required samples, a small shovel was used to carefully separate the roots from the remaining soil until the root structure was completely exposed. The number and angle (*θ*) of the roots were measured and recorded. The roots were projected onto a vertical whiteboard with pre-drawn directions and lines. The numbers of horizontal roots (0° < *θ* ≤ 30°), oblique roots (30° < *θ* ≤ 60°), and vertical roots (60° < *θ* ≤ 90°) were manually counted and recorded. After recording, all roots within the soil layer were cut into individual segments using scissors. The length of each individual root (*RL*) was measured with a ruler to an accuracy of 0.01 mm. The diameter of each individual root at the top, middle, and end was measured with a vernier caliper to an accuracy of 0.01 mm, and the average value was taken as the root diameter (*RD*). Based on the root diameter (*d*), the roots were divided into four size categories: fine roots (*d* < 2 mm), small roots (2 ≤ *d* ≤ 5 mm), medium roots (5 ≤ *d* ≤ 10 mm), and coarse roots (*d* ≥ 10 mm) [10]. Assuming the roots are cylindrical, the root surface area (*RSA*) and root volume (*RV*) were calculated based on the measured *RD* and *RL* [6,12].
(1)$$RSA=\sum_{i}^{n}\pi \cdot RD_{i}\cdot RL_{i}$$
(2)$$RV=\frac{\sum_{i}^{n}\pi \cdot \left(\frac{RD_{i}}{2}\right)\cdot RL_{i}}{V}$$
where *RSA* and *RV* are the root surface area and root volume, respectively, *RD_i_* is the diameter (mm) of the *i*-th root in the sample, *RL_i_* is the length (mm) of the *i*-th root, and *V* is the soil layer volume.

After recording the root data, plant roots from different soil layers were dried at 65 °C for 48 h to calculate the root mass density (*RMD*, kg∙m^−3^), root length density (*RLD*, km∙m^−3^), root surface area density (*RSAD*, m^2^∙m^−3^), and root volume density (*RVD*, m^3^∙m^−3^) [6,33].
(3)$$RMD=\frac{\sum_{i}^{n}RM_{i}}{V}$$
(4)$$RLD=\frac{\sum_{i}^{n}RL_{i}}{V}$$
(5)$$RSAD=\frac{\sum_{i}^{n}RSA_{i}}{V}$$
(6)$$RVD=\frac{\sum_{i}^{n}RV_{i}}{V}$$

### 2.3. In Situ Shear Tests and Shear Strength Indices

This study utilized a self-designed and custom-fabricated in situ shear test apparatus. The dimensions of the instrument frame were 150 cm × 35 cm × 16 cm, with a shear box size of 20 cm × 20 cm × 10 cm and a shear plane area of 0.04 m^2^. The instrument connected the shear box to the guide rail using clamps and sliders. To ensure a stable shear direction, three 60 cm long iron rods were used to secure the instrument at the front and rear into the soil. Pressure sensors (0–3000 ± 0.1 N) and displacement sensors (0–200 ± 0.1 mm) were used for data measurement and recording. The shear rate was set at 0.3 mm/s [18].

Before the shear test began, a rectangular trench was excavated to accommodate the shear apparatus, with a reserved root-containing soil volume of 25 cm × 25 cm × 10 cm. The shear apparatus was placed in the trench, aligning the reserved root-containing soil with the shear box, and minor adjustments were made to the surrounding soil outside the shear box. The instrument was fixed in place using iron stakes, and the sensors were zeroed before commencing the test. The specific experimental procedure is shown in Figure 2.

For each vegetation type, 6 plants were selected, and shear tests were conducted on soil layers at depths of 0–10, 10–20, 20–30, 30–40, and 40–50 cm, with 6 tests conducted per layer. The average values of these tests were used to create the shear stress–displacement curves for each soil layer. Additionally, shear tests were performed on wasteland using the same method as for the soil–root systems. Three representative parameters (strain energy *SE*, peak shear stress *PSS*, and peak shear stress–displacement *DPS*) were obtained from the shear stress–displacement curves to reflect the shear resistance of the soil–root systems. The *PSS* and *DPS* were determined from the peak points on the curve. The *SE* was calculated as the integral of the curve over the range (0, *DPS*).

The in situ shear tests provided shear stress–displacement curves for different plants in the 0–50 cm soil layers. The *PSS* values for the six plants and the wasteland were obtained for each soil layer. The difference between the *PSS* values of each plant and the wasteland in each layer, denoted as the shear strength increment (Δ*PSS*), represented the increased shear resistance of the soil–root systems due to the presence of roots.

### 2.4. Wu–Waldron Model and Root Tensile Tests

#### 2.4.1. Root Tensile Test

To avoid physical damage to the roots in each soil layer caused by shear tests, six additional plants of each vegetation type were selected. Using a stratified and segmented excavation method, the roots of the six plants were obtained from the 0–10, 10–20, 20–30, 30–40, and 40–50 cm soil layers. After obtaining complete root systems, individual roots that were straight, had minimal diameter variation, no surface damage, and a length of 100 mm were cut for sample preparation. The diameters of the roots at both ends and the middle were measured using a vernier caliper with an accuracy of 0.01 mm [9,19].

The single root tensile test was conducted using a universal testing machine (China-Sansi-UTM5105, Shanghai, China) to determine the tensile force and tensile strength of roots of different diameters from the six plant species. The samples were clamped on both ends with the clamps of the universal testing machine. To prevent damage to the roots from the clamps and to provide better grip, a thin layer of paper was placed between the clamps and the roots. The clamps moved at a constant speed of 10 mm∙min^−1^, applying tensile force to the roots. Only samples that broke in the middle third of the length between the clamps were considered valid, with a test success rate calculated as between 30% and 55%. To minimize environmental effects on the roots, all tests were completed within 12 h of obtaining them.

#### 2.4.2. Construction of the Wu–Waldron Model

The Wu model posits that the role of plant roots in enhancing slope soil primarily lies in root reinforcement and anchorage. Plant roots combine with the soil to form an integrated system that enhances soil strength by providing additional cohesion [34]. The foundation of this model is the assumption that all roots intersecting the shear plane break during the shearing process [13,35]. The model states that the shear strength of root-reinforced soil can be calculated using the Mohr–Coulomb equation, as follows [13]:(7)$$\tau_{sr}=C_{s}+C_{r}+\sigma \tan\phi$$
where *C_s_* is the soil cohesion, *C_r_* is the additional cohesion due to the presence of roots, *σ* is the normal stress on the shear plane of the soil–root system (kPa), and *ϕ* is the internal friction angle of the soil (°). Assuming the roots are elastic and the Mohr–Coulomb criterion is effective for shear [9], and that the internal friction angle of the soil is unaffected, the additional root cohesion *C_r_* can be estimated as [5,13]
(8)$$C_{r}=t_{r}\left(\sin\delta +\cos\delta \tan\phi \right)$$
where *δ* is the angle of the roots undergoing shear deformation relative to the shear plane, and *t_r_* is the product of the average tensile strength of the roots and the root area ratio (*RAR*), which is the fraction of the soil area occupied by roots. The value of $\left(sin\delta +cos\delta tan\phi \right)$ can be approximated as 1.2 [9,13]. Therefore, the equation can be rewritten as
(9)$$C_{r}=1.2\cdot T_{r}\cdot RAR$$
where *T_r_* is the average tensile strength of the roots per unit soil area, obtained from single root tensile tests. RAR is the ratio of the total root area (*A_r_*) to the soil area (*A_s_*) for each soil layer. *C_r_* is the additional cohesion due to the presence of roots.

### 2.5. Data Analysis

One-way analysis of variance (ANOVA) and Tamhane T2 post hoc test were used to analyze the differences in soil properties (*BD*, *SWC*, and *SOM*), root traits (*RLD*, *RMD*, *RSAD*, *RSVD*, *CRR*, *MR*, *FR*, *HR*, *OR*, and *VR*), and shear strength parameters (SE, PSS, and DPS) of six plant types in the 0–50 cm soil layers (*p* < 0.05). Regression analysis was employed to establish the relationship between root tensile strength and diameter, as well as to analyze the specific relationship between shear strength parameters, soil properties, and root traits. Principal component analysis (PCA) was used to group different vegetation types and soil depths, analyzing the correlations between shear strength parameters, soil properties, and root traits to determine the relationships between various variables. All statistical analyses were performed using SPSS 26.0 and Origin Pro 21.0 software.

## 3. Results

### 3.1. Root Traits and Soil Properties of Different Vegetation Types

As the soil depth increases, the organic matter content gradually decreases, while the soil moisture content and bulk density both gradually increase. Regarding the soil moisture content, except for *P. sepium*, which shows a decreasing trend along the layers, the moisture content of the wasteland and the other six plant–soil–root systems shows an increasing trend. For the bulk density, as the soil depth increases, the bulk density of the six soil–root systems and the wasteland all show an increasing trend, with the wasteland exhibiting more significant changes compared to the other six soil–root systems. Within the same soil layer, the bulk density of the wasteland is the highest, while that of *A. hedinii* and *P. sepium* is lower. Regarding changes within the same soil layer, the decrease in the organic matter content is more significant compared to the changes in the bulk density and soil moisture content (Figure A1).

The roots of the two shrubs, *Z. jujuba* and *P. sepium*, are longer and distributed throughout the 0–50 cm soil layers, whereas the roots of the other four plant species are only distributed within the 0–30 cm soil layers. As the soil depth increases, the root length density, root mass density, root surface area density, and root volume density all show a clear decreasing trend. The distribution of the root mass density and root volume density is relatively consistent across the soil layers (Figure A2). The roots of the six plant species are mainly composed of fine roots, with very few coarse and small roots, accounting for no more than 5%. *A. hedinii* has the largest number of roots, followed by *B. papyrifera* and *S. viridis*, while *P. sepium* and *Z. jujuba* have the fewest (Figure A3a,b). *S. viridis* and *B. papyrifera* are dominated by vertical roots, with *B. papyrifera* containing a small number of horizontal and oblique roots, and *S. viridis* almost entirely lacking horizontal and oblique roots. *P. przewalskii* is primarily composed of horizontal roots, with some oblique and vertical roots. *A. hedinii* has the most roots, mainly consisting of oblique roots (Figure A4a,b).

For different vegetation types, the soil properties and root traits of the 0–50 cm soil profile are significantly affected by the vegetation. Specific data can be found in the Appendix A (Table A1, Table A2 and Table A3).

### 3.2. In Situ Shear Stress–Displacement Curves and Shear Strength

The shear stress–displacement curves for the different vegetation types show significant differences. Among them, *Z. jujuba*, *B. papyrifera*, and *P. przewalskii* have higher shear stress–displacement curves in the 0–10 cm and 10–20 cm soil layers. Overall, the wasteland has the lowest shear stress–displacement curves (Figure 3g), followed by *S. viridis* and *P. sepium* (Figure 3b,c). The shear stress–displacement curves of *P. przewalskii*, *B. papyrifera*, and *Z. jujuba* are relatively higher and steeper, with more pronounced peak points. For the two herbaceous plants, *S.viridis* and *A.hedinii*, only the 0–10 cm soil layer curve height is significantly higher than in other soil layers, with the peak points also significantly skewed to the right. The curves of the other soil layers for the two herbaceous plants are similar in height, with little difference (Figure 3a,b). For the shrubs, the curves of *Z.jujuba* are clearly higher than those of *P.sepium* (Figure 3c,d). For the two tree species, the curves of *B.papyrifera* in the 0–10 cm, 10–20 cm, and 20–30 cm soil layers are significantly higher than those in the 30–40 cm and 40–50 cm soil layers. The curves of *P.przewalskii* in the 0–10 cm and 10–20 cm soil layers are significantly higher than in the other three soil layers (Figure 3e,f). The curves of the different soil layers in the wasteland are more concentrated compared to the soil layers with vegetation (Figure 3g).

As the soil depth increases, the *SE* of the six slope protection vegetation types gradually decreases. Compared to wasteland, the *PSS* of *A. hedinii* and *S. viridis* significantly increases only in the 0–10 cm soil layer, with no significant changes in the other four layers. For *P. sepium*, *Z. jujuba*, *P. przewalskii*, and *B. papyrifera*, the *PSS* in all five soil layers is significantly higher than in the wasteland. The *DPS* fluctuates significantly with the soil depth, but the *DPS* of almost all the vegetation types shows a decreasing trend as the soil depth increases. The *SE*, *PSS*, and *DPS* of the six plant types differ from those of the wasteland in each soil layer, but the average strain energy (*SE_m_*) of *Z. jujuba*, *B. papyrifera*, *A.hedinii*, and *P. przewalskii* shows even greater differences compared to those of the wasteland (Table 2). Compared to the wasteland, the *SE_m_* of *A. hedinii*, *S. viridis*, *P. sepium*, *Z. jujuba*, *P. przewalskii*, and *B. papyrifera* in the 0–30 cm soil layer increased by 61.65 J·m^2^, 23.22 J·m^2^, 54.86 J·m^2^, 118.96 J·m^2^, 87.27 J·m^2^, and 117.66 J·m^2^, respectively.

### 3.3. Root Tensile Strength

There are significant differences in the measured tensile strength of roots between different vegetation types, and the tensile strength within the same species is strongly influenced by the root diameter. The relationship between the root diameter and tensile strength was fitted using a regression equation (Figure 4a–f). The tensile strength decreases with increasing root diameter. The range of tensile strengths is as follows: *P. sepium* ranges from 15.98 MPa to 6.07 MPa, *Z. jujuba* ranges from 44.86 kPa to 8.93 kPa, *P. przewalskii* ranges from 38.93 kPa to 8.28 kPa, *B. papyrifera* ranges from 68.25 kPa to 7.46 kPa, *A. hedinii* ranges from 56.62 kPa to 11.34 kPa, and *S. viridis* ranges from 95.08 kPa to 53.67 kPa.

### 3.4. Comprehensive Analysis of Shear Strength of Soil–Root Systems

The roots of all six plants significantly enhanced the shear strength of the soil–root systems. The soil–root system of *A. hedinii* in the shallow layer (0–10 cm) had the largest shear strength increment, at 1370.54 kPa. The shear strength increments of *B. papyrifera* and *P. przewalskii* in each soil layer were also significant and noticeably higher than those of *P. sepium* and *S. viridis* (Table 3). Since the vast majority of roots are distributed in the 0–30 cm soil layer, the average additional cohesion (*C_r_*) for the 0-30 cm layer was calculated based on the results of the Wu model. The ranking of the average additional cohesion (*C_rm_*) is as follows: *A. hedinii* > *B. papyrifera* > *P. przewalskii* > *Z. jujuba* > *P. sepium* > *S. viridis*. The shear strength increments obtained from the in situ shear tests for the six vegetation types were ranked as follows: *Z. jujuba* > *B. papyrifera* > *P. przewalskii* > *A. hedinii* > *P. sepium* > *S. viridis*.

### 3.5. Shear Strength, Soil Properties, and Root Traits

The soil properties and root traits are closely related to the shear strength parameters of the soil–root system (Figure 5, Table 4). *SE* is negatively correlated with *SWC* and positively correlated with *RMD*, *MR*, and *RVD* (*p* < 0.05). *PSS* is positively correlated with *MR*, *RMD*, *RSAD*, *FR*, and *RLD*, and negatively correlated with *SWC* (*p* < 0.05). *DPS* is positively correlated with *MR*, *RMD*, *RVD*, *ASD*, *FR*, and *RLD*, and negatively correlated with *SWC* and *BD* (*p* < 0.05). Additionally, there is a strong correlation between the three parameters *SE*, *PSS*, and *DPS* (*p* < 0.01). The three shear strength indicators have a low correlation with *HR*, *OR*, and *VR*, indicating that these indicators are poorly related to the spatial morphology of the roots. The three indicators of shear strength all show a strong correlation with *RMD* and *SWC*. *SE*, *PSS*, and *DPS* all increase with the power function of *RMD* and decrease with the power function of *SWC*, with determination coefficients (*R*^2^) of 0.28, 0.46, 0.49, 0.27, 0.23, and 0.37, respectively. Additionally, *DPS* increases with the power function of *RVD* and *FR*, with determination coefficients (*R*^2^) of 0.33 and 0.14, respectively (Figure 6a–h).

## 4. Discussion

The shear strength of soil–root systems is one of the most effective dynamic indicators for assessing soil structure stability, as it reflects the ability of the soil–root system to withstand shear forces and deformation pressures [36,37]. The differences in the root traits among the different vegetation types can significantly affect the soil properties, which in turn influence the shear strength of soil–root systems. In this study, the root traits of the same plant type were similar across different soil layers, likely due to the similar sampling environments and consistent standards used when selecting test plants. This consistency indicates stable root development, representative of normal plant growth during the study period. Research on root traits shows that, overall, plant roots are more abundant in shallow soil layers, with the number of roots decreasing as the soil depth increases [9]. This decrease in root number with increasing soil depth may be due to reduced soil aeration, decreased soil fertility, and more compacted soil at greater depths [38]. In the study of root traits for the six plants, significant differences were observed in the root parameters among the different plant species. These differences are mainly determined by the genetic characteristics of the tree species under the same growth conditions and are also influenced by the soil slope, tree age, and climate [14].

From the perspective of soil reinforcement and slope protection, fine roots have a stronger reinforcing effect on soil. This study found that roots with diameters less than 2 mm are the main contributors to the total root number and root length density, consistent with previous research [10]. The roots of *A. hedinii* have abundant fine roots, significantly reinforcing the soil. However, the roots of *Z. jujuba* have fewer fine roots but still show considerable reinforcing effects, possibly due to the hardness and high tensile strength of *Z. jujuba* roots. Soil reinforcement mechanisms are influenced by the tensile properties of roots [39,40,41]. In this study, the root diameter of the six plant species was significantly negatively correlated with the tensile strength, with a power function relationship between the root tensile properties and root diameter, similar to that seen in previous findings. The similarities and differences in the tensile strength–diameter relationship among the different plant roots reflect the internal and external structural characteristics of the roots [42].

This study used both in situ shear tests and the Wu model to measure and estimate the increase in the shear strength of soil–root systems due to the presence of roots. For in situ shear tests, the shear strength indices are represented by three parameters: *SE*, *PSS*, and *DPS* [6,12,15,21,30]. The Wu model quantifies the slope reinforcement effect of different plant roots by calculating the additional cohesion in the soil due to the presence of roots [13].

There is a certain degree of discrepancy between the shear strength increments of soil–root systems obtained using the Wu model and those obtained through in situ shear tests, with the Wu model generally yielding larger increments (Table 3). This difference may be due to the assumption in the Wu model that all roots break simultaneously during the shearing of the soil–root system. In reality, the roots do not break simultaneously but fail progressively, with some roots gradually breaking and others being pulled out directly. This leads to the shear strength increment due to roots being higher than the measured value, thereby overestimating the reinforcement effect of roots. However, the overall trend in the results from the Wu model is consistent with those from the in situ shear tests, with *A. hedinii*, *B. papyrifera*, *P. przewalskii*, and *Z. jujuba* all showing a good slope reinforcement capability using both methods. In the 0–10 cm soil layer, the shear strength increment of *A. hedinii* obtained from the Wu model is significantly higher than that from the in situ shear test, likely due to the very dense and developed shallow root system of *A. hedinii*. In the in situ shear tests, *Z. jujuba* showed the best slope reinforcement effect, but its effect was lower than that of *P. przewalskii* and *B. papyrifera* in the Wu model. This could be because *Z. jujuba* roots were found to be thicker and harder during the root survey, but they were fewer in number, putting *Z. jujuba* at a disadvantage in the Wu model calculations (Table 3).

The shape of the shear stress–displacement curves obtained from the in situ shear tests illustrates the mechanical behavior of the soil–root systems during the shear tests [16]. Overall, the curve for the fallow land is the lowest, while the curves for *S. viridis* and *P. sepium* are relatively flat. In contrast, the shear stress–displacement curves for *Z. jujuba*, *B. papyrifera*, and *P. przewalskii* are steeper, with more pronounced peak points. This phenomenon is attributed not only to the differences in the root systems of the six plants but also to differences in the soil properties [6]. For the same plant species, the shear stress–displacement curves of the different soil layers show significant differences. The curves for the 0–30 cm soil layers are steeper and have peak points further to the upper right compared to those for the 30–50 cm soil layers. Taking *A. hedinii* as an example, the shear stress–displacement curves for the 0–10 cm and 10–20 cm soil layers are very steep, while those for the 20–30 cm, 30–40 cm, and 40–50 cm soil layers are relatively flat. The main reason for this difference is that *A. hedinii* has a well-developed root system, with a large main root diameter and numerous lateral roots, most of which are distributed in the 0–20 cm soil layers.

Future research should investigate the shear strength of soil–root systems for more plant species, various tree ages, and greater soil depths, as well as their relationships with the climate, soil type, and topography. Additionally, this study only considered the mechanical effects of plant roots for plant selection, without addressing deeper landslide mechanisms and economic considerations.

## 5. Conclusions

This study investigated the shear strength of soil–root systems of six vegetation types in the Yellow River Basin of China using both the Wu–Waldron model and in situ shear tests. It explored the relationship between the shear strength, soil properties, and root traits. Significant differences were observed in the root traits among the six woodland, shrubland, and grassland plant species. As the soil depth increased, the *RLD*, *RMD*, *RSAD*, and *RVD* showed a clear decreasing trend. Additionally, there were significant variations in the soil properties; specifically, *SOM* gradually decreased, while *SWC* and *BD* both increased with soil depth. The soil properties and root traits significantly influenced the shear strength of the soil–root systems. In the in situ shear tests, the three shear strength indices—*SE*, *PSS,* and *DPS*—were positively correlated with *RMD* and negatively correlated with the soil water content (*SWC*). The results from both the in situ shear tests and the Wu–Waldron model were consistent. Considering both methods, *Ziziphus jujuba*, *Populus przewalskii*, *Broussonetia papyrifera*, and *Artemisia hedinii* demonstrated the best slope protection effects in shallow soil layers. These findings may provide guidance for selecting vegetation for slope protection efforts in the Yellow River Basin.

## Figures and Tables

**Figure 1 plants-13-02963-f001:**
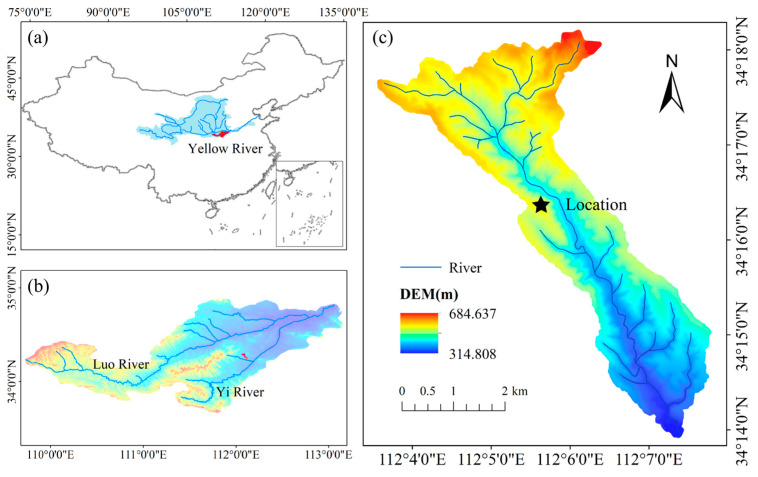
Location of Hugou Basin and the experimental site. (**a**) The Yellow River Basin in China. (**b**) The Yiluo River Basin. (**c**) The Hugou Basin.

**Figure 2 plants-13-02963-f002:**
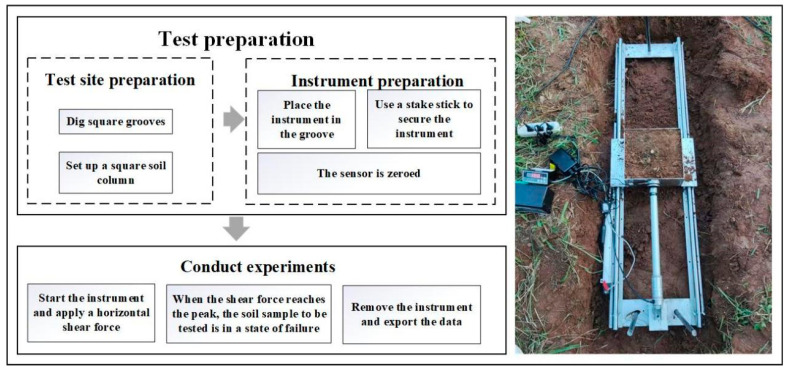
Procedure for in situ shear test.

**Figure 3 plants-13-02963-f003:**
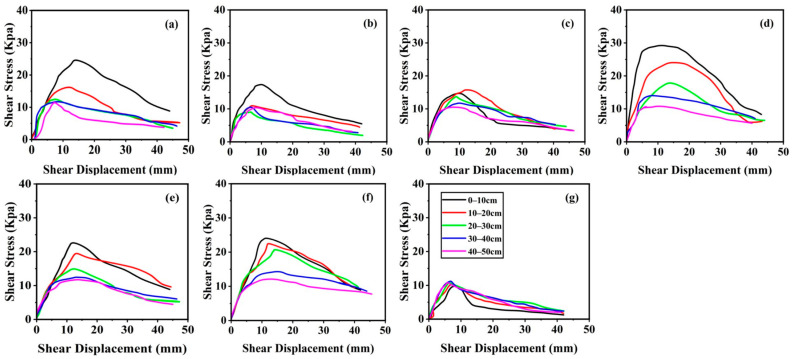
Shear stress–displacement curves of different soil layers under various vegetation types. Source: author’s own work. Figures (**a**–**g**) correspond to *Artemisia hedinii* (*A. hedinii*), *Setaria viridis* (*S. viridis*), *Periploca sepium* (*P. sepium*), *Ziziphus jujuba* (*Z. jujuba*), *Populus przewalskii* (*P. przewalskii*), *Broussonetia papyrifera* (*B. papyrifera*), and wasteland, respectively.

**Figure 4 plants-13-02963-f004:**
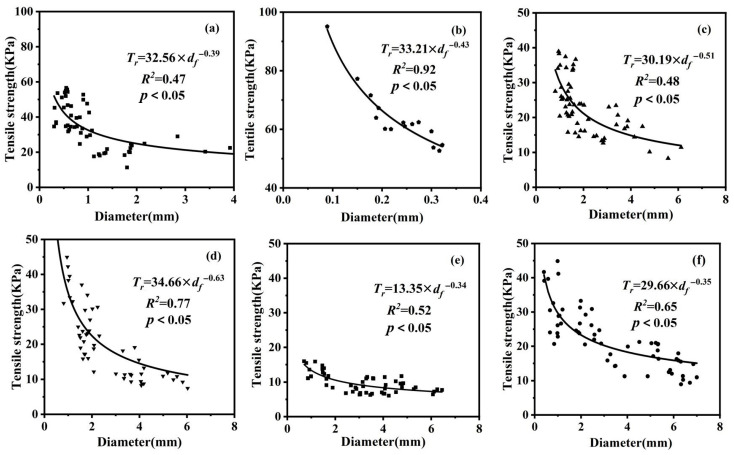
Power regression curves showing the relationship between tensile strength and root diameter. Source: author’s own work. (**a**–**f**) correspond to *Artemisia hedinii* (*A. hedinii*), *Setaria viridis* (*S. viridis*), *Periploca sepium* (*P. sepium*), *Ziziphus jujuba* (*Z. jujuba*), *Populus przewalskii* (*P. przewalskii*), and *Broussonetia papyrifera* (*B. papyrifera*), respectively.

**Figure 5 plants-13-02963-f005:**
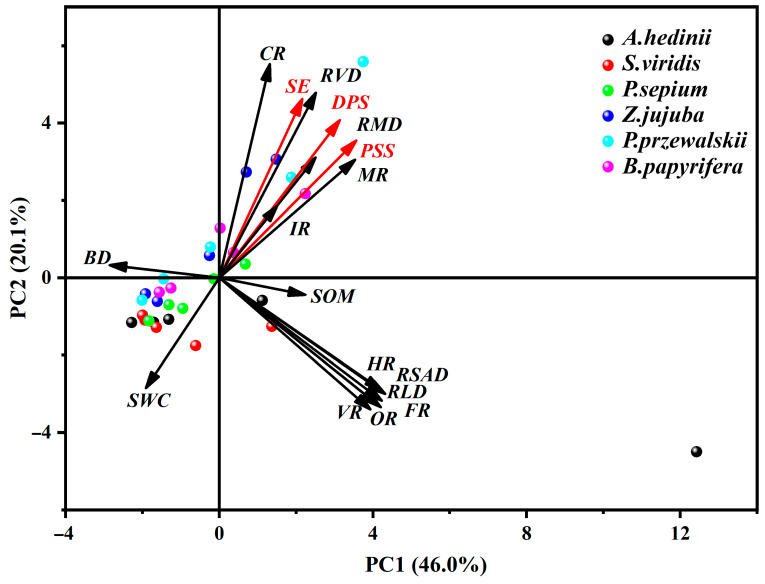
Principal component analysis results of shear strength parameters, soil properties, and root morphology. *PSS* is peak shear stress, *DPS* is peak shear stress–displacement. *BD* is soil bulk density, *SOM* is soil organic matter content, *SWC* is soil water content. *RLD* is root length density, *RMD* is root mass density, *RSAD* is root surface area density, *RVD* is root volume density. *FR* is fine roots, *MR* is small roots, *IR* is medium roots, *CR* is coarse roots. *HR* is horizontal roots, *OR* is oblique roots, *VR* is vertical roots.

**Figure 6 plants-13-02963-f006:**
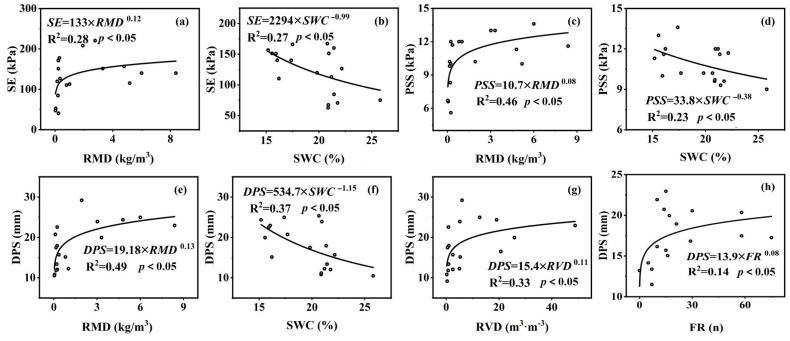
Relationships between shear strength indicators, soil properties, and root traits. (**a**,**b**) Relationship between *SE* and *RMD*, *SWC*. (**c**,**d**) Relationship between *PSS* and *RMD*, *SWC*. (**e**–**h**) Relationship between *DPS* and *RMD*, *SWC*, *RVD*, *FR*.

**Table 1 plants-13-02963-t001:** Characteristics of experimental sites and vegetation.

Land Use	Plant Species	Aspect	Slope (%)	Age (year)	Height (cm)	Canopy Width (cm)	Number (n)
Wasteland	/	Sunny	10	/	/	/	/
Grassland	*A. hedinii*	Sunny	10	Annual herb	70.67 ± 3.30	41.33 ± 3.09	30
	*S. viridis*	Sunny	10	Annual herb	59.33 ± 1.70	44.17 ± 2.39	30
Shrubland	*P. sepium*	Sunny	10	2	123.67 ± 2.62	25.67 ± 0.47	28
	*Z. jujuba*	Sunny	10	2	124.00 ± 4.32	70.67 ± 4.19	29
Woodland	*P. przewalskii*	Sunny	10	2	158.67 ± 4.64	23.50 ± 0.71	25
	*B. papyrifera*	Sunny	10	2	99.67 ± 6.02	24.50 ± 6.75	24

**Table 2 plants-13-02963-t002:** Strain energy (*SE*, J·m^2^) and peak shear stress (*PSS*, kPa) for different vegetation types.

Plant Species	0–10 cm		10–20 cm		20–30 cm		30–40 cm		40–50 cm		Mean	
	*SE*	*PSS*	*SE*	*PSS*	*SE*	*PSS*	*SE*	*PSS*	*SE*	*PSS*	*SE_m_*	*PSS_m_*
Wastseland	49.40	9.82	48.39	10.3	38.39	10.97	47.33	11.4	38.22	10.98	44.35	10.69
*A. hedinii*	206.05	24.95	148.97	16.40	65.42	12.55	70.13	11.75	39.45	11.65	106	15.46
*S. viridis*	129.02	17.48	55.45	11.08	48.68	10.15	49.28	10.8	55.43	10.53	67.57	12.01
*P. sepium*	112.07	14.78	151.98	15.7	78.65	13.83	84.43	11.8	68.90	10.53	99.21	13.33
*Z. jujuba*	248.39	29.23	227.13	24.12	171.14	17.93	78.50	14.6	91.40	10.83	163.31	19.34
*P. przewalskii*	139.02	22.95	153.66	19.95	121.53	15.15	122.18	12.53	121.69	11.78	131.62	16.47
*B. papyrifera*	152.78	24.35	155.46	22.55	209.03	20.7	157.21	14.38	135.59	12.25	162.01	18.85

Note: Since the majority of the roots of the six plant species are distributed within the 0–30 cm soil layer, the average strain energy of the 0–30 cm soil layer is denoted as the average strain energy (*SE_m_*). The average peak shear stress of the 0–30 cm soil layer is denoted as *PSS_m_*.

**Table 3 plants-13-02963-t003:** Additional root cohesion (*C_r_*, kPa) from the Wu model and shear strength increment (Δ*PSS*, kPa) from in situ shear tests.

Plant Species	0–10 cm		10–20 cm		20–30 cm		30–40 cm		40–50 cm		Mean	
	*Cr*	∆*PSS*	*Cr*	∆*PSS*	*Cr*	∆*PSS*	*Cr*	∆*PSS*	*Cr*	∆*PSS*	*Cr_m_*	∆*PSS_m_*
*A. hedinii*	1370.54	15.13	224.35	6.10	46.17	2.45	0	0.35	0	1.45	547.02	7.89
*S. viridis*	63.45	7.66	35.58	0.78	15.41	0.05	0	−0.60	0	0.33	38.15	2.83
*P. sepium*	98.49	4.96	80.94	5.40	6.12	3.73	3.06	0.40	3.06	0.33	61.85	4.70
*Z. jujuba*	73.80	19.41	169.39	13.82	50.06	7.83	18.16	2.66	0	0.63	97.75	13.69
*P. przewalskii*	224.52	13.13	82.59	9.65	25.73	5.05	0	1.13	0	1.58	110.95	9.28
*B. papyrifera*	238.99	14.53	75.54	12.25	30.18	10.63	0	2.98	0	2.05	114.90	12.47

Note: Since the majority of the roots of the six plant species are distributed within the 0–30 cm soil layer, the average additional root cohesion for the 0–30 cm soil layer is denoted as *C_rm_*. The average shear strength increment for the 0–30 cm soil layer is denoted as Δ*PSS_m_*.

**Table 4 plants-13-02963-t004:** Correlation coefficients between shear strength indicators, soil properties, and root traits.

	*SE*	*PSS*	*DPS*	*SOM*	*SWC*	*BD*	*RLD*	*RMD*	*ASD*	*RVD*	*FR*
*PSS*	0.78 **										
*DPS*	0.89 **	0.65 **									
*SOM*	0.11	0.01	0.25								
*SWC*	−0.39 *	−0.52 *	−0.55 **	−0.03							
*BD*	−0.26	−0.20	−0.37 *	−0.56 **	−0.01						
*RLD*	0.16	0.31 *	0.35 *	0.48 *	−0.18	−0.51 **					
*RMD*	0.39 *	0.36 *	0.57 **	0.36 *	−0.32 *	−0.41 *	0.45 **				
*ASD*	0.17	0.33 *	0.39 *	0.35 *	−0.24	−0.48 **	0.96 **	0.49 **			
*RVD*	0.28	0.30	0.45 **	0.31 *	−0.38 *	−0.30	0.20	0.91 **	0.21		
*FR*	0.15	0.31 *	0.36 *	0.34 *	−0.23	−0.23 *	0.96 **	0.44 **	0.99 **	0.16	

Note: * *p* < 0.05, ** *p* < 0.01.

## Data Availability

All data and materials used in this manuscript are freely available and comply with field standards. The data presented in this study are available upon request from the corresponding author.

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
