# Peer review of "Study on Shear Strength of Soil–Root Systems of Different Vegetation Types"

_plants, 2024, doi:10.3390/plants13212963_

Round 1
Reviewer 1 Report
Comments and Suggestions for Authors
Article ID plants-3249682, entitled “Study on Shear Strength of Soil-Root Systems of Different Vegetation Types” submitted to the Journal Plants.
This is an experimental study on the evaluation of soils on slopes, based on the soil-root system.
The study has merits and relevance, is well written and organized, but needs to improve some aspects that hinder a better appreciation of the content.
Words and expressions from the title should be avoided in the keywords.
Citations between brackets in superscript should be corrected.
Line 56: correct the citation to: Xing et al. (2023a)
Line 59: unnumbered citation that corresponds to reference [33], which needs to be corrected and which will change the numbering of all subsequent citations in the article.
Line 65: correct the citation to: De Baets et al. (2008)
Line 66: correct the citation to: Su et al. (2021)
Line 77: correct the citation to: Reubens et al. (2007)
Improve the layout of the figure, clearly identifying the basin where the experimental site was installed
Line 150: correct the citation: (Zhang et al., 2024), which is unnumbered. [10]????
Improve the expression of the equations, which should follow a uniform standard of expression and layout.
I recommend a more detailed description of the shear test apparatus, for the purpose of reproducing the experiment by interested readers.
Lines 232 and 233: Unreferenced citations
Carefully check the callouts of Figures A1, A2, A3, and A4 and Tables A1-13, which do not correspond to the citations in the text.
Figure 4: Explain the plant species corresponding to the letters a, b, c, d, e, and f.
Authors should be careful with very low R2s, especially those below 0.6, in Figures 4 and 6, which do not allow for a discussion with the necessary statistical support.
Review the conclusion that seems to be a continuation of the discussions.
There are also other observations in the attached text

Reviewer 2 Report
Comments and Suggestions for Authors
SWC (soil water content) - it is advisable to explain and justify in which units it is given (% volume or % weight).
Reviewer 3 Report
Comments and Suggestions for Authors
Dear Authors,
please find enclosed the comments on your manuscript.
Regards
Reviewer

The quality of English is fine.
Reviewer 4 Report
Comments and Suggestions for Authors
The study is important for the development of knowledge in the field
Provides insight into future research in the field
Figures A1, A2, A3, A4 and tables A1-A3 are mentioned in the paper, but they are not found in the paper, which makes it difficult to evaluate the interpretation of the results
Round 2
Reviewer 1 Report
Comments and Suggestions for Authors
The article was carefully reviewed by the authors, who complied with the previously recommended adjustment requests.
In the attached text I make two suggestions for corrections.

Reviewer 4 Report
Comments and Suggestions for Authors
The authors have made substantial additions to the work in response to the observations made
The citation in the text of figure 1 (line 123) to which of the figures it refers (fig 1a or fig 1 a-c)?
Line 330 figure 4 - which figure is referred to? (figs 4 a-f or just fig 4.b?)
Line 371 figure 6 - which of the figures is referred to? (figures 6a-h or only fig. 1.c?, 1g?)
